# Comparison of the accuracy of diagnoses of oral potentially malignant disorders with dysplasia by a general dental clinician and a specialist using the Taiwanese Nationwide Oral Mucosal Screening Program

Tien-En Chiang[1,2], Yu-Chun Lin[3], Chi-Tsung Wu[1,2], Cheng-Yu Yang[2], Sheng-Tang Wu[4,5☯], Yuan-Wu Chen [1,2☯] *

1 Division of Oral and Maxillofacial Surgery, Tri-Service General Hospital, Taipei, Taiwan, R.O.C, 2 School of Dentistry, National Defense Medical Center, Taipei, Taiwan, R.O.C, 3 Department of Pathology, Tri-Service General Hospital, National Defense Medical Center, Taipei, Taiwan, R.O.C, 4 Division of Urology, Department of Surgery, Tri-Service General Hospital, National Defense Medical Center, Taipei, Taiwan, R.O. C, 5 Department of Medical Planning, Medical Affairs Bureau, Ministry of National Defense, Taipei, Taiwan, R.O.C

☯ These authors contributed equally to this work.
* h6183@yahoo.com.tw

## Abstract

Screening for oral potentially malignant disorders (OPMDs) with dysplasia in high-risk groups is suggested in countries with a high prevalence of the disorders. This study aimed to compare the accuracy of diagnoses of OPMDs with dysplasia made by a primary examiner (general dental clinician) and a specialist (oral and maxillofacial surgeon) using the current Taiwanese Nationwide Oral Mucosal Screening Program (TNOMSP). A total of 134 high-risk participants were enrolled for oral mucosal screening via the TNOMSP. A primary examiner and a specialist examined each participant. Mucosal biopsies were obtained and subjected to histopathological analysis. The OPMD most frequently diagnosed by the primary examiner was thin homogeneous leukoplakia (48/134; 35.8%), and in 39/134 participants (29.1%) the diagnosis was uncertain, but abnormalities were suggested. The OPMDs most frequently diagnosed by the specialist were erythroleukoplakia (23/134; 17.2%) and thin homogeneous leukoplakia (21/134; 15.7%), and 51/134 participants (38.1%) were diagnosed with other diseases. Via histopathology, 70/134 participants (52.3%) were diagnosed with dysplasia, and 58/134 (43.3%) were diagnosed with benign conditions. The specialist's diagnoses exhibited a higher specificity, positive predictive value, and accuracy than the primary examiners. A specialist using the current TNOMSP for high-risk participants diagnosed OPMDs with dysplasia more accurately than a primary examiner. Early diagnosis of high-risk OPMDs is crucial in countries with a high prevalence of the disorders. Proficient examination via the current TNOMSP by trained clinician is effective for the management of OPMDs with dysplasia.

**Data Availability Statement:** All relevant data are within the manuscript and its Supporting Information files.

**Funding:** This work was supported by Tri-Service General Hospital, Taiwan, R.O.C. (grant numbers TSGH-C106-004-006-008-S05, TSGH-C107-008-S06, TSGH-C108-007-008-S06, TSGH C01-109017, and TSGH-D-109163), the Taiwan Ministry of Science and Technology (grant numbers MOST 105-2314-B-016-021-MY3 and MOST 108-2314-B-016-005).

**Competing interests:** The authors have declared that no competing interests exist.

## Introduction

Several terms have been used to refer to the mucosal pathology that develops prior to oral cancer, such as "pre-cancerous", "precancerous/premalignant lesions", and "intraepithelial neoplasia" [1–3]. Warnakulasuriya et al. [4] proposed the more precise term–oral potentially malignant disorders (OPMDs)–to refer to conditions such as leukoplakia, erythroplakia, and submucous fibrosis, and stated that they are a family of morphological alterations with the potential for malignant transformation. The spectrum of OPMDs includes oral leukoplakia, erythroplakia, erythroleukoplakia, oral submucous fibrosis, palatal lesions in reverse smokers, oral lichen planus, oral lichenoid reactions, and other disorders involving systemic disease or hereditary conditions [5].

The histopathological diagnosis of OPMDs varies and can include hyperplasia, hyperkeratosis, and oral epithelial dysplasia (OED) [6]. OEDs can be categorized as mild, moderate, or high based on the presence and severity of cell atypia and structural epithelial involvement [7]. The OED grade has predictive value with regard to the potential for malignancy in individuals with oral mucosal lesions or OPMDs [8]; reported associated rates of malignant transformation range from 2.2% to 38.1% [9,10].

Risk factors for OPMDs differ by country and region, and in Taiwan, the most common OPMDs are leukoplakia, erythroplakia, and oral submucous fibrosis [11–13]. The majority of these disorders may be asymptomatic in the early years but can be detected by dental clinicians during routine oral examinations [14]. It is crucial for health professionals to be well educated on the clinical features and diagnostic aspects of OPMDs, when to further investigate, and when it is appropriate to refer the patient to a specialist for treatment. The current treatment protocol for OPMDs is based on the morphology of the lesion and the stage of dysplasia of the lesion. For mild dysplasia, the treatment can range from observation to excision based on the clinician's judgment, but moderate to high-grade dysplasia generally requires surgical intervention [15]. Therefore, the precision of conventional oral examination and diagnosis of OED is crucial in terms of prognosis and malignant transformation [16].

One of the aims of conventional oral mucosa screening is early diagnosis of potentially malignant lesions, thus reducing malignancy-related mortality. Screening for OPMDs can facilitate preventive intervention [17], and it is relatively easy to conduct in a stepwise manner to identify lesions [18]. In Taiwan, the Taiwanese Nationwide Oral Mucosal Screening Program (TNOMSP) has been supported by the government for many years [19]. It incorporates a specific checklist that can be used by general dental clinicians or other health professionals while conducting an initial examination [20], but a more conclusive diagnosis is determined by a specialist and histopathology. After diagnosis, a follow-up for recall and monitoring is instigated in high-risk patients with treated lesions [21]. As the criteria for the specialist required by TNOMSP include management, the diagnosis of OPMDs is crucial. Therefore, the accuracy of diagnoses of OPMDs with dysplasia by a primary examiner (general dental clinician) and a specialist (oral and maxillofacial surgeon) based on the TNOMSP were compared, with reference to histopathology results.

## Materials and methods

### Patient recruitment

The study was approved by the Institutional Review Board of the Tri-Service General Hospital of National Defense Medical Center, Taipei, Taiwan (IRB: 1-107-05-010) and was conducted within the Department of Oral and Maxillofacial Surgery at that hospital. The inclusion criteria were age > 20 years, a history of tobacco and/or betel nut use (*i.e.*, high-risk individuals), and

being of Han Chinese descent. A total of 150 consecutive patients were initially screened for inclusion in the study, of which 16 were subsequently excluded; 10 because they declined to participate and 6 because they were not of Han Chinese descent.

## Oral mucosal screening

An oral mucosal screening program checklist developed by the Health Promotion Administration at the Ministry of Health and Welfare is commonly used for oral cancer screening as well as screening for OPMDs in Taiwan. The checklist includes basic information about the participants, betel nut and tobacco use, and awareness of the lesion, and it was used to assess all patients in the current study.

## Clinical examination procedure

The clinical examination checklist from TNOMSP was used. A diagnosis using the TNOMSP checklist was performed by a primary examiner (general dental clinician), who recorded clinical diagnoses of erythroplakia, erythroleukoplakia, verrucous hyperplasia, non-homogeneous leukoplakia, homogeneous thick leukoplakia, submucosal fibrosis, lichen planus, unknown mass, unhealed ulceration for > 2 weeks (Fig 1), abnormal mucosa without a diagnosis, suspected oral cancer, and other conditions. The same patient went through a clinical examination performed by a certified specialist examiner (oral and maxillofacial surgeon) who clinically diagnosed non-homogeneous leukoplakia, homogeneous thick leukoplakia, leukoplakia, erythroplakia, erythroleukoplakia, verrucous hyperplasia, submucosal fibrosis, lichen planus, suspected oral cancer, and other conditions. Mucosal biopsies of all lesions were performed and subjected to histopathological analysis (Fig 2).

Diagnoses were made using the Taiwanese Nationwide Oral Mucosal Screening Program checklist. The primary examiner suspected a diagnosis of oral cancer but categorized the diagnosis into an unconfirmed diagnosis of an unhealed ulceration for > 2 weeks in the checklist.

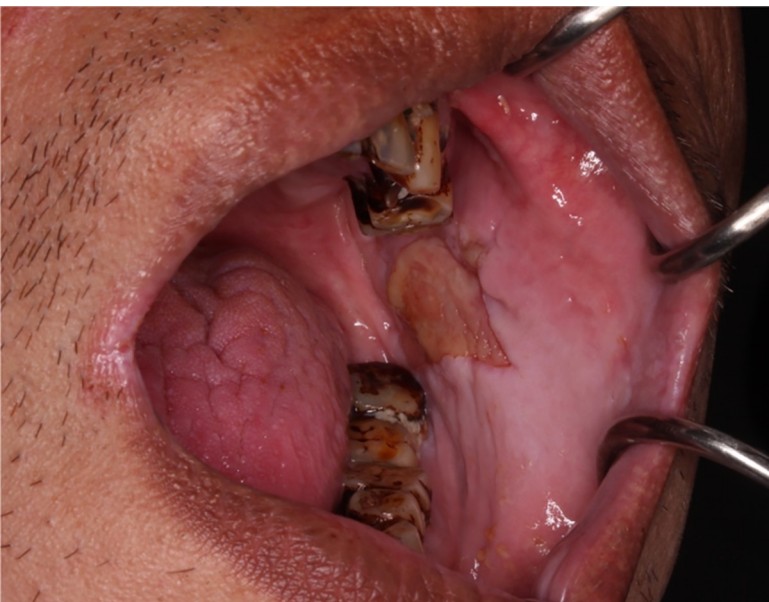

**Fig 1. The patient shown exhibited an oral ulcerative lesion on the left buccal mucosa at the retromolar region that the patient had been aware of for 2 weeks, and betel nut staining and tooth attrition were noted in the dentition.**

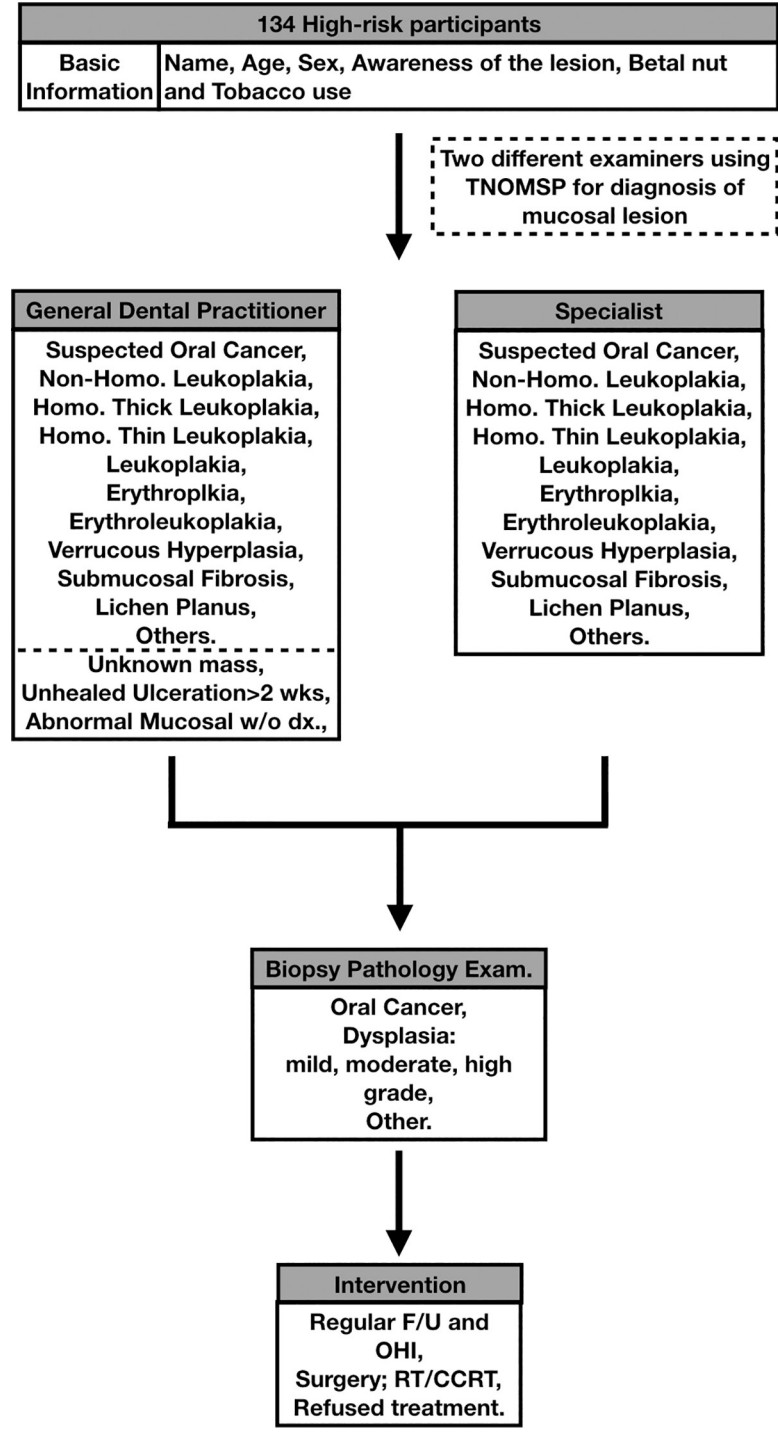

**Fig 2. Oral mucosal screening protocol.**

The specialist categorized the lesion as "other." Biopsy histopathology indicated inflammation without dysplasia. The patient was advised to stop chewing betel nuts, rounding of the tooth attrition was performed, and a follow-up visit was arranged.

### Histopathological assessment

Patients signed a standard informed consent form that is typically used at the hospital. A biopsy was performed for histopathological assessment, and the biopsy site was selected from the diagnosed lesion site. The presence or absence of dysplasia, oral cancer, or other abnormalities in the biopsy specimen was recorded in the pathology report and approved by a certified pathologist.

### Statistical analysis

All statistical analyses were performed using the statistical software SPSS (version 22.0.0, IBM Corp., Armonk, NY, USA). Responses were numerically coded to facilitate data entry. The McNemar's test was used to determine the difference between primary and the specialist in diagnosing OPMDs. The sensitivity, specificity, positive predictive value, negative predictive value, and accuracy of the clinical diagnoses made by the primary examiner and the specialist examiner were determined based on the gradings derived from the biopsies. The receiver operating characteristic (ROC) curve and area under the ROC curve were used to assess the accuracy of diagnoses of dysplasia and cancer made by the primary examiner and the specialist. In all analyses, $p < 0.05$ (two-tailed) was considered statistically significant.

## Results

Application of the inclusion and exclusion criteria resulted in 134/150 (89.3%) initially screened patients being included in the study; 117 men and 17 women with a mean age of 56.55 ± 12.93 years. Most participants (114/134; 85.1%) were unaware of their lesions. A history of chewing betel nuts was reported by 63.5% of participants, and current smoking or a history of smoking was reported by 76.8%. A total of 76/134 participants (56.7%) were diagnosed with dysplasia and/or cancer via histopathology, and 58/134 (43.2%) were diagnosed with benign conditions (Table 1).

### Clinical diagnoses made by the primary examiner and the specialist

The results of the conventional oral examinations are shown in Table 2. The OPMD diagnosed most frequently by the primary examiner was thin homogeneous leukoplakia (48/134; 35.8%), and in 39/134 (29.1%) the diagnosis was uncertain, but an abnormality was suggested. The OPMD diagnosed most frequently by the specialist was erythroleukoplakia (23/134; 17.2%), followed by thin homogeneous leukoplakia (21/134; 15.7%); 51/134 (38.1%) participants were diagnosed with other diseases.

### Accuracy of diagnoses of OPMDs with dysplasia by the primary examiner and the specialist

The results of statistical evaluations performed to assess associations between clinical and histopathological diagnoses and examiner experience are shown in Table 3. Clinical diagnoses of OPMDs made by the specialist exhibited a higher specificity, positive predictive value, and accuracy with reference to histopathological diagnoses. The overall accuracy of diagnoses of dysplasia made by the specialist was 85.4%, compared to 62.5% for the primary examiner, and with ROC curve analysis the difference was found to be significant (Fig 3).

## Discussion

Identifying OED associated with OPMDs is crucial because of the potential for malignant transformation. OED is classified into three stages: mild, moderate, and severe. In mild OED,

**Table 1. Patient characteristics and histopathological diagnoses.**

| Variable | All Patients N = 134 | Primary[a] | | | Specialist | | | Histopathology | | |
|---|---|---|---|---|---|---|---|---|---|---|
| | | OPMDs N = 82 | Cancer N = 5 | Other N = 8 | OPMDs N = 73 | Cancer N = 10 | Other N = 51 | Dysplasia N = 70 | Cancer N = 6 | Other N = 58 |
| Age | 56.55 ± 12.93 | 56.28 ± 11.10 | 56.67 ± 8.49 | 40.99 ± 21.46 | 56.69 ± 10.24 | 61.11 ± 12.19 | 55.47 ± 16.17 | 58.71 ± 11.77 | 62.45 ± 13.07 | 53.34 ± 13.70 |
| **Sex** | | | | | | | | | | |
| Female | 17 (12.7%) | 8 (9.8%) | 0 (0%) | 0 (0%) | 10 (13.7%) | 1 (10.0%) | 6 (11.8%) | 13 (18.6%) | 1 (16.7%) | 3 (5.2%) |
| Male | 117 (87.3%) | 74 (90.2%) | 5 (100%) | 8 (100%) | 63 (86.3%) | 9 (90.0%) | 45 (88.2%) | 57 (81.4%) | 5 (83.3%) | 55 (94.8%) |
| **Awareness of the lesion** | | | | | | | | | | |
| No | 114 (85.1%) | 74 (90.2%) | 3 (60.0%) | 4 (50.0%) | 65 (89.0%) | 2 (20.0%) | 47 (92.2%) | 63 (90.0%) | 0 (0%) | 51 (87.9%) |
| Yes | 20 (14.9%) | 8 (9.8%) | 2 (40.0%) | 4 (50.0%) | 8 (11.0%) | 8 (80.0%) | 4 (7.8%) | 7 (10.0%) | 6 (100%) | 7 (12.1%) |
| **Betel nut use** | | | | | | | | | | |
| Never used | 49 (36.6%) | 27 (32.9%) | 1 (20.0%) | 4 (50.0%) | 31 (42.5%) | 2 (20.0%) | 16 (31.4%) | 27 (38.6%) | 2 (33.3%) | 20 (34.5%) |
| Ex-user | 36 (26.9%) | 27 (32.9%) | 1 (20.0%) | 4 (50.0%) | 15 (20.5%) | 1 (10.0%) | 20 (39.2%) | 12 (17.1%) | 2 (33.3%) | 22 (37.9%) |
| Current user | 49 (36.6%) | 28 (34.1%) | 3 (60.0%) | 0 (0%) | 27 (37.0%) | 7 (70.0%) | 15 (29.4%) | 31 (44.3%) | 2 (33.3%) | 16 (27.6%) |
| **Tobacco use** | | | | | | | | | | |
| Never smoked | 31 (23.1%) | 14 (17.1%) | 1 (20.0%) | 4 (50.0%) | 14 (19.2%) | 1 (10.0%) | 16 (31.4%) | 15 (21.4%) | 1 (16.7%) | 15 (25.9%) |
| Ex-smoker | 31 (23.1%) | 21 (25.6%) | 1 (20.0%) | 4 (50.0%) | 19 (26.0%) | 1 (10.0%) | 11 (21.6%) | 12 (17.1%) | 2 (33.3%) | 17 (29.3%) |
| Current smoker | 72 (53.7%) | 47 (57.3%) | 3 (60.0%) | 0 (0%) | 40 (54.8%) | 8 (80.0%) | 24 (47.1%) | 43 (61.4%) | 3 (50.0%) | 26 (44.8%) |

a: Participants diagnosed with an unexplained persistent mass, unhealed ulceration for more than 2 weeks, or abnormal mucosa without diagnosis by primary examiner were excluded.

Data are presented as the mean ± standard deviation.

**Table 2. Clinical diagnoses made by the primary examiner and the specialist using the Taiwanese Nationwide Oral Mucosal Screening Program.**

| Variable | Primary N = 134 | Specialist N = 134 | p |
|---|---|---|---|
| **Suspected oral cancer** | 5 (3.7%) | 10 (7.5%) | < 0.001 |
| **Oral potentially malignant disorders** | | | |
| Erythroplakia | 4 (3.0%) | 5 (3.7%) | |
| Erythroleukoplakia | 15 (11.2%) | 23 (17.2%) | |
| Non-homogeneous leukoplakia | 5 (3.7%) | 7 (5.2%) | |
| Thick homogeneous leukoplakia | 3 (2.2%) | 5 (3.7%) | |
| Thin homogeneous leukoplakia | 48 (35.8%) | 21 (15.7%) | |
| Verrucous hyperplasia | 2 (1.5%) | 6 (4.5%) | |
| Submucous fibrosis | 2 (1.5%) | 3 (2.2%) | |
| Lichen planus | 3 (2.2%) | 3 (2.2%) | |
| Unexplained persistent mass | 2 (1.5%) | NA | |
| Unhealed ulceration for > 2 weeks | 17 (12.7%) | NA | |
| Abnormal mucosa without diagnosis | 20 (14.9%) | NA | |
| Other | 8 (6.0%) | 51 (38.1%) | |

p values were calculated using the McNemar's test.

NA, not applicable in the Taiwanese Nationwide Oral Mucosal Screening Program checklist.

**Table 3. Accuracy of diagnoses of oral potentially malignant disorders with dysplasia by the primary examiner and the specialist.** Accuracy of diagnoses of oral potentially malignant disorders with dysplasia by the specialist.

| Patients exclude due to cancer N = 88 | | Primary examiner | |
|---|---|---|---|
| | | OPMDs[a] N = 80 | Other N = 8 |
| Pathology report | Dysplasia N = 47 | 47 (53.4%) | 0 (0%) |
| | Other N = 41 | 33 (37.5%) | 8 (9.1%) |
| Sensitivity | | 100 | |
| Specificity | | 19.5 | |
| Positive predictive value | | 58.8 | |
| Negative predictive value | | 100 | |
| Accuracy | | 62.5 | |
| AUC | | 0.598 | |
| Patients exclude due to cancer N = 123 | | Specialist | |
| | | OPMDs[a] N = 72 | Other N = 51 |
| Pathology report | Dysplasia N = 66 | 60 (48.8%) | 6 (4.9%) |
| | Other N = 57 | 12 (9.8%) | 45 (36.6%) |
| Sensitivity | | 90.9 | |
| Specificity | | 78.9 | |
| Positive predictive value | | 83.3 | |
| Negative predictive value | | 88.2 | |
| Accuracy | | 85.4 | |
| AUC | | 0.870[b] | |

a: Oral potentially malignant disorders (OPMDs) include erythroplakia, erythroleukoplakia, non-homogeneous leukoplakia, thick homogeneous leukoplakia, thin homogeneous leukoplakia, verrucous hyperplasia, submucous fibrosis, and lichen planus.

b: log-rank test compared with AUC = 0.5, p < 0.001

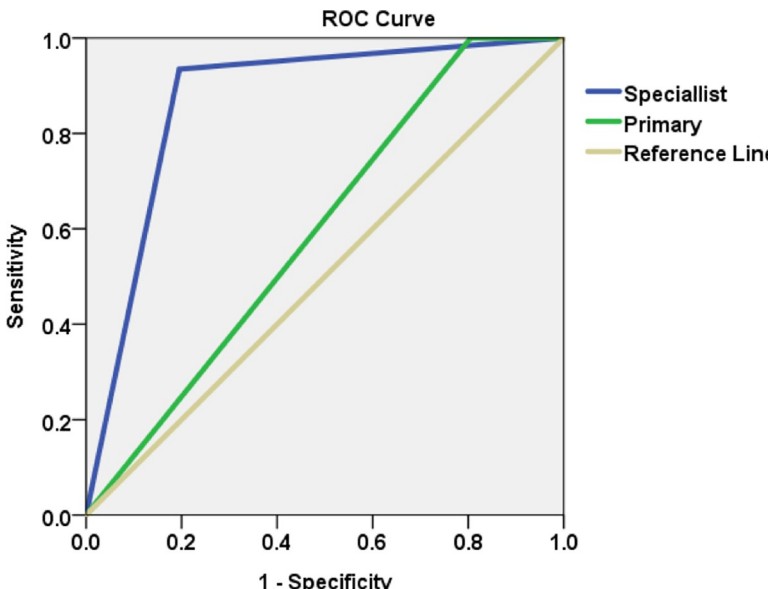

**Fig 3. Receiver operating characteristic curve for comparison of the diagnostic accuracy of the primary examiner and the specialist.** The area under curve (AUC) for the specialist (blue) and primary examiner (green) were 0.870 and 0.598, respectively; log-rank test comparing the specialist and primary examiner was significant (p < 0.001).

dysplastic changes are limited to basal or parabasal layers; in moderate OED, there is involvement of the middle or granular layer, and in severe OED, changes are evident from the basal layer of the epithelium to the middle and upper layers [22]. The current protocol in our department is to administer aggressive treatment for moderate- and high-grade dysplasia. This is because high-grade epithelial dysplasia has an overall malignant transformation rate of approximately 16%; though, reported rates vary widely, from 7% to 50% [23]. Moderate dysplasia has reported rates of malignant transformation ranging from 15.0% to 26.8%, and mild epithelial dysplasia is reportedly associated with a rate of malignant transformation of < 5% [24–26]. Regardless of its classification, at our institution, all patients diagnosed with OED are strongly urged to attend all scheduled follow-up visits.

In the current study, we included high-risk individuals screened separately by both examiners based on the original design of TNOMSP. The TNOMSP was initiated in 1985, and thereafter it was gradually scaled up to include all of Taiwan, and to specifically target high-risk individuals. Unlike western countries, the incidence of oral mucosal malignant changes is approximately one-hundred times higher in high-risk individuals than the general population due to betel nut and tobacco use [19,20,27]. The TNOMPS has been promoted by government health administrators and broadcast by media to highlight the importance of mucosal screening, which has been implanted for decades. Funding has been distributed through the Taiwan national health insurance system to first-line public health care providers, including dentists in general practice, oral and maxillofacial surgeons, otolaryngologists, and other related physicians. High-risk individuals are targeted for screening biennially, including those who currently use or have quit the use of betel nut and tobacco, as well as those above the age of 30. In 2015, there were 939,000 screenings conducted, and the percentage of the screened high-risk individuals rose from 28% in 2009 to 56.1% in 2015; the above-mentioned screening detected OPMDs in 4095 individuals and oral malignancy in 1361 individuals [20,28]. Several international studies indicate that para-habit users are at high risk of OPMDs [29–31]. In Southeast Asia, approximately 90% of deaths resulting from oral malignancy occur in patients with habits known to be associated with enhanced risks of those malignancies; thus, the specific allocation of resources to screen individuals in high-risk groups is justifiable [32]. Due to the often-asymptomatic onset of OPMDs, the majority of patients with OED are evidently unaware of their condition, as was the case in the current study; thus, educating high-risk groups about their risks and encouraging them to engage in the TNOMSP may improve prevention rates [33].

The primary aim of the TNOMSP is early diagnosis of OPMDs and consequent prevention of progression to malignant transformation. Accordingly, appropriate education to facilitate the identification of such lesions is crucial [34]. Understandably, there are substantial differences in the levels of training and experience between general dental clinicians and specialists (e.g. oral and maxillofacial surgeons) regarding the accurate diagnosis of oral diseases. In two studies undertaken to investigate referrals after oral mucosal screening, a lack of confidence due to insufficient training was suggested in the context of general dental clinicians making accurate diagnoses [35,36]. This was also observed in the current study, in which the general dental clinician tended to make non-specific diagnoses such as unexplained persistent mass, unhealed ulceration for > 2 weeks, and abnormal mucosa without a definitive diagnosis. Notably however, in some countries, there is a lack of professional medical resources, and the presence or absence of a mucosal lesion may serve as an indication of whether further investigations or referral to a specialist are required [37].

Patients with OPMDs are referred for biopsy and treated, and in abnormal cases, close follow-up by a specialist is recommended. In a recent large-scale retrospective cohort study conducted in Taiwan investigating associations between OPMDs and the early diagnosis of oral

cancer, an oral mucosal screening program for the early identification of OPMDs contributed substantially to reducing mortality [34]. Interestingly, OSF is a unique OPMDs in South-East Asia due to para-habits. A large national scale 10-year observation study conducted in Taiwan for OPMDs, targeting oral submucous fibrosis (OSF) with oral leukoplakia (OL), revealed a higher and faster malignant transformation rate then OSF alone [38]. Another study was conducted in a high-occurrence region within a single hospital in southern Taiwan, in which 555 individuals with OPMDs during a 5 year-period were examined for the relationship of its clinicopathological features and transformation. In this study, the annual malignant transformation rate was 1.16%, specifically for OSF, and that for homogenous leukoplakia and non-homogenous leukoplakia was 5.7%, 4.6%, and 12.1%, respectively [39]. A well-designed surveillance program could lead to early malignancy detection and reduced mortality and morbidity. This has been also demonstrated in a large scale cohort study using the Taiwan Cancer Registry to evaluate the relationship between OPMDs and malignancy, which showed a survival benefit due to early diagnosis of OPMDs to prevent oral malignancies [34]. For many years, this program has benefited the high-risk group, and there has been a 21% reduction in stage III or IV oral cancer diagnoses, and a 26% reduction in oral cancer mortality [40,41].

## Conclusions

In the current study, a clinical specialist, such as an oral and maxillofacial surgeon well trained and prepared to make a correct diagnosis for OPMDs, demonstrated a better outcome in diagnosing histopathologic OED. Our findings also suggest that, based on the design of the TNOMSP, the high-risk group can be properly diagnosed by the specialist for treatment. However, the limitation of this study is small sample size in a single medical center with limited examiners, but this provides a rationale for further comparative evaluation of the TNOMSP in diagnosis of OPMDs with dysplasia by more qualified examiners or specialists in a larger population base study.

## Supporting information

**S1 Data.**
(PDF)

## Acknowledgments

The authors acknowledge the services provided the staffs of Division of Oral and Maxillofacial Surgery, Tri-Service General Hospital, National Defense Medical Center.

## Author Contributions

**Data curation:** Tien-En Chiang, Yu-Chun Lin.

**Formal analysis:** Tien-En Chiang.

**Methodology:** Cheng-Yu Yang.

**Project administration:** Sheng-Tang Wu, Yuan-Wu Chen.

**Resources:** Chi-Tsung Wu.

**Writing – original draft:** Tien-En Chiang.

**Writing – review & editing:** Sheng-Tang Wu, Yuan-Wu Chen.

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
