## [Decision Letter · Decision Letter 0]

14 Oct 2020

PONE-D-20-25840

Comparison of the accuracy of diagnoses of oral potentially malignant disorders with dysplasia by a general dental clinician and a specialist using the Taiwanese Nationwide Oral Mucosal Screening Program

PLOS ONE

Dear Dr. Chen

Thank you for submitting your manuscript to PLOS ONE. After careful consideration, we feel that it has merit but does not fully meet PLOS ONE’s publication criteria as it currently stands. Therefore, we invite you to submit a revised version of the manuscript that addresses the points raised during the review process.

We look forward to receiving your revised manuscript.

Kind regards,

Cheng-Chia Yu 

Academic Editor

PLOS ONE

Journal Requirements:

Reviewers' comments:

Reviewer's Responses to Questions

**Comments to the Author**

1. Is the manuscript technically sound, and do the data support the conclusions?

Reviewer #1: Partly

Reviewer #2: Partly

2. Has the statistical analysis been performed appropriately and rigorously? 

Reviewer #1: No

Reviewer #2: Yes

3. Have the authors made all data underlying the findings in their manuscript fully available?

Reviewer #1: No

Reviewer #2: Yes

4. Is the manuscript presented in an intelligible fashion and written in standard English?

Reviewer #1: Yes

Reviewer #2: Yes

5. Review Comments to the Author

Reviewer #1: 1.As you mention in the conculsion, a clinical specialist such as an oral and maxillofacial surgeon is well

251 trained and prepared on making a correct diagnosis for OPMDs, which also revealed a better

252 outcome in finding histopathologic OED.

It is alreday a static and patient choosen bias. Since this is not a RCT, in clinical condition, symptomatic patient may go or refer to specialist already. It's not fair to compare the two group under this circumstances.

2.Table 1. Patient characteristics and histopathological diagnoses should also be seperate into two different comparsion group.

3.Table 3. should aslo present the real case number rather than the percentage only.

4.What is the outcome of Screening for oral potentially malignant disorders (OPMDs) in Tawain and other hospital should be mention insie the article.

5.Discussion part should be more organized and corrlated to the topic you want to discussion about. What kind of patient or lesion should be performed the screen should be mentioned inside discussion.

6.The limitation of the study showed also be mentioned.

Reviewer #2: In the present paper, the authors investigate the accuracy of diagnoses of OPMDs with dysplasia using Taiwanese Nationwide Oral 6 Mucosal Screening Program. Very interesting work but from reading the abstract and methods section, I was expecting to see some more proposed about the statistical analysis which was claimed by the author. And there are some parts of the research needed to be deliberate.

1. What is the criterion for the receiver operating characteristic (ROC)?

2.The author should deliberate about the description of using the receiver operating characteristic (ROC) curve for sensitivity, specificity, positive predictive value, negative predictive value, and accuracy .Cause there is not enough evidence to support the description of statistical analysis.

6. PLOS authors have the option to publish the peer review history of their article (what does this mean?). If published, this will include your full peer review and any attached files.

Reviewer #1: No

Reviewer #2: No

---

## [Author Response · Author response to Decision Letter 0]

25 Nov 2020

Responses by the Author

1. Is the manuscript technically sound, and do the data support the conclusions?

Reviewer #1: Partly

Reviewer #2: Partly

2. Has the statistical analysis been performed appropriately and rigorously? 

Reviewer #1: No

Reviewer #2: Yes

3. Have the authors made all data underlying the findings in their manuscript fully available?

Reviewer #1: No

Reviewer #2: Yes

Response: We appreciate Reviewer 1 for reminding us of the importance of making our original data open access. We have hereby uploaded our original data for the purpose of science.

4. Is the manuscript presented in an intelligible fashion and written in standard English?

Reviewer #1: Yes

Reviewer #2: Yes

5. Review Comments to the Author

Reviewer #1: 1.As you mention in the conculsion, a clinical specialist such as an oral and maxillofacial surgeon is well 

251 trained and prepared on making a correct diagnosis for OPMDs, which also revealed a better

252 outcome in finding histopathologic OED.

It is alreday a static and patient choosen bias. Since this is not a RCT, in clinical condition, symptomatic patient may go or refer to specialist already. It's not fair to compare the two group under this circumstances.

Response: We apologize for the misguiding figures and the text in the manuscript. This is a comparative study using paired data. We have revised our manuscript and figures to make them more understandable as follows:

Fig 2. Oral mucosal screening protocol.

Pg 2.

Abstract

Early diagnosis of high-risk OPMDs is crucial in countries with a high prevalence. Stepwise examination via the current TNOMSP and appropriate subsequent referral is effective for the management of OPMDs with dysplasia. A specialist using the current TNOMSP for high-risk participants diagnosed OPMDs with dysplasia more accurately than a primary examiner. Early diagnosis of high-risk OPMDs is crucial in countries with a high prevalence of the disorders. Proficient examination via the current TNOMSP by trained clinician is effective for the management of OPMDs with dysplasia.

Pg. 5

Clinical examination procedure

The clinical examination checklist from procedure for the TNOMSP was used. Primary oral mucosal screening A diagnosis using the TNOMSP checklist was performed by a primary examiner (general dental clinician), who recorded clinical diagnoses of erythroplakia, erythroleukoplakia, verrucous hyperplasia, non homogeneous leukoplakia, homogeneous thick leukoplakia, submucosal fibrosis, lichen planus, unknown mass, unhealed ulceration for > 2 weeks (Fig 1), abnormal mucosa without a diagnosis, suspected oral cancer, and other conditions. The same patient went through a A conclusive clinical examination was performed by a certified specialist examiner (oral and maxillofacial surgeon) who clinically diagnosed non-homogeneous leukoplakia, homogeneous thick leukoplakia, leukoplakia, erythroplakia, erythroleukoplakia, verrucous hyperplasia, submucosal fibrosis, lichen planus, suspected oral cancer, and other conditions. Mucosal biopsies of all lesions were performed and subjected to histopathological analysis (Fig 2).

2.Table 1. Patient characteristics and histopathological diagnoses should also be seperate into two different comparsion group.

Response: We thank reviewer 1 for this insightful suggestion; we have using paired data and revised our table 1 as following: 

Pg. 8

Table 1. Patient characteristics and histopathological diagnoses

Variable All Patients

N = 134 Primarya Specialist Histopathology

 OPMDs

N = 82 Cancer

N = 5 Other

N = 8 OPMDs

N = 73 Cancer

N = 10 Other

N = 51 Dysplasia

N = 70 Cancer

N = 6 Other

N = 58

Age 56.55 ± 12.93 56.28 ± 11.10 56.67 ± 8.49 40.99 ± 21.46 56.69 ± 10.24 61.11 ± 12.19 55.47 ± 16.17 58.71 ± 11.77 62.45 ± 13.07 53.34 ± 13.70

Sex 

Female 17 (12.7%) 8 (9.8%) 0 (0%) 0 (0%) 10 (13.7%) 1 (10.0%) 6 (11.8%) 13 (18.6%) 1 (16.7%) 3 (5.2%)

Male 117 (87.3%) 74 (90.2%) 5 (100%) 8 (100%) 63 (86.3%) 9 (90.0%) 45 (88.2%) 57 (81.4%) 5 (83.3%) 55 (94.8%)

Awareness of the lesion 

No 114 (85.1%) 74 (90.2%) 3 (60.0%) 4 (50.0%) 65 (89.0%) 2 (20.0%) 47 (92.2%) 63 (90.0%) 0 (0%) 51 (87.9%)

Yes 20 (14.9%) 8 (9.8%) 2 (40.0%) 4 (50.0%) 8 (11.0%) 8 (80.0%) 4 (7.8%) 7 (10.0%) 6 (100%) 7 (12.1%)

Betel nut use 

Never used 49 (36.6%) 27 (32.9%) 1 (20.0%) 4 (50.0%) 31 (42.5%) 2 (20.0%) 16 (31.4%) 27 (38.6%) 2 (33.3%) 20 (34.5%)

Ex-user 36 (26.9%) 27 (32.9%) 1 (20.0%) 4 (50.0%) 15 (20.5%) 1 (10.0%) 20 (39.2%) 12 (17.1%) 2 (33.3%) 22 (37.9%)

Current user 49 (36.6%) 28 (34.1%) 3 (60.0%) 0 (0%) 27 (37.0%) 7 (70.0%) 15 (29.4%) 31 (44.3%) 2 (33.3%) 16 (27.6%)

Tobacco use 

Never smoked 31 (23.1%) 14 (17.1%) 1 (20.0%) 4 (50.0%) 14 (19.2%) 1 (10.0%) 16 (31.4%) 15 (21.4%) 1 (16.7%) 15 (25.9%)

Ex-smoker 31 (23.1%) 21 (25.6%) 1 (20.0%) 4 (50.0%) 19 (26.0%) 1 (10.0%) 11 (21.6%) 12 (17.1%) 2 (33.3%) 17 (29.3%)

Current smoker 72(53.7%) 47 (57.3%) 3 (60.0%) 0 (0%) 40 (54.8%) 8 (80.0%) 24 (47.1%) 43(61.4%) 3 (50.0%) 26 (44.8%)

a: participants diagnosed with an unexplained persistent mass, unhealed ulceration for more than 2 weeks, or abnormal mucosa without diagnosis by primary examiner were excluded. 

Data are presented as the mean ± standard deviation 

3.Table 3. should aslo present the real case number rather than the percentage only.

Response: We thank Reviewer 1 for this comment; We have revised the table as follows:

Pg. 11

Table 3-1. Accuracy of diagnoses of oral potentially malignant disorders with dysplasia by the primary examiner and the specialist

Patients exclude due to cancer

N = 88 Primary examiner

 OPMDsa

N = 80 Other

N = 8

Pathology report Dysplasia

N = 47 47 (53.4%) 0 (0%)

 Other

N = 41 33 (37.5%) 8 (9.1%)

Sensitivity 100

Specificity 19.5

Positive predictive value 58.8

Negative predictive value 100

Accuracy 62.5

AUC 0.598

a: Oral potentially malignant disorders (OPMDs) include erythroplakia, erythroleukoplakia, non-homogeneous leukoplakia, thick homogeneous leukoplakia, thin homogeneous leukoplakia, verrucous hyperplasia, submucous fibrosis, and lichen planus.

Table 3-2. Accuracy of diagnoses of oral potentially malignant disorders with dysplasia by the specialist

Patients exclude due to cancer

N = 123 Specialist

 OPMDsa

N = 72 Other

N = 51

Pathology report Dysplasia

N = 66 60 (48.8%) 6 (4.9%)

 Other

N = 57 12 (9.8%) 45 (36.6%)

Sensitivity 90.9

Specificity 78.9

Positive predictive value 83.3

Negative predictive value 88.2

Accuracy 85.4

AUC 0.870b

a: Oral potentially malignant disorders (OPMDs) include erythroplakia, erythroleukoplakia, non-homogeneous leukoplakia, thick homogeneous leukoplakia, thin homogeneous leukoplakia, verrucous hyperplasia, submucous fibrosis, and lichen planus, 

b: log-rank test compared with AUC=0.5, p < 0.001

4.What is the outcome of Screening for oral potentially malignant disorders (OPMDs) in Tawain and other hospital should be mention insie the article.

Response: We thank reviewer 1 for helping us improve our manuscript. We have revised our discussion as follows:

Pg. 15

Discussion

Patients with OPMDs are referred for biopsy and treated, and in abnormal cases, close follow-up by a specialist is recommended. In a recent large-scale retrospective cohort study conducted in Taiwan investigating associations between OPMDs and the early diagnosis of oral cancer, an oral mucosal screening program for the early identification of OPMDs contributed substantially to reducing mortality [34]. Interestingly, OSF is a unique OPMDs in South-East Asia due to para-habits. A large national scale 10-year observation study conducted in Taiwan for OPMDs, targeting oral submucous fibrosis (OSF) with oral leukoplakia (OL), revealed a higher and faster malignant transformation rate then OSF alone [35]. Another study was conducted in a high-occurrence region within a single hospital in southern Taiwan, in which 555 individuals with OPMDs during a 5 year-period were examined for the relationship of its clinicopathological features and transformation. In this study, the annual malignant transformation rate was 1.16%, specifically for OSF, and that for homogenous leukoplakia and non-homogenous leukoplakia was 5.7%, 4.6%, and 12.1%, respectively [36]. A well-designed surveillance program could lead to early malignancy detection and reduced mortality and morbidity. This has been also demonstrated in a large scale cohort study using the Taiwan Cancer Registry to evaluate the relationship between OPMDs and malignancy, which showed a survival benefit due to early diagnosis of OPMDs to prevent oral malignancies [34]. For many years, this program has benefited the high-risk group, and there has been a 21% reduction in stage III or IV oral cancer diagnoses, and a 26% reduction in oral cancer mortality [40,41].

5.Discussion part should be more organized and corrlated to the topic you want to discussion about. What kind of patient or lesion should be performed the screen should be mentioned inside discussion.

Response: We thank reviewer 1 for helping us improve our manuscript. We have revised our discussion as follows:

Pg. 13.

Discussion

In the current study, we included high-risk individuals screened separately by both examiners based on the original design of TNOMSP. The TNOMSP was initiated in 1985, and thereafter it was gradually scaled up to include all of Taiwan, and to specifically target high-risk individuals. Unlike western countries, the incidence of oral mucosal malignant changes is approximately one-hundred times higher in high-risk individuals than the general population due to betel nut and tobacco use [19-21]. The TNOMPS has been promoted by government health administrators and broadcast by media to highlight the importance of mucosal screening, which has been implanted for decades. Funding has been distributed through the Taiwan national health insurance system to first-line public health care providers, including dentists in general practice, oral and maxillofacial surgeons, otolaryngologists, and other related physicians. High-risk individuals are targeted for screening biennially, including those who currently use or have quit the use of betel nut and tobacco, as well as those above the age of 30. In 2015, there were 939,000 screenings conducted, and the percentage of the screened high-risk individuals rose from 28% in 2009 to 56.1% in 2015; the above-mentioned screening detected OPMDs in 4095 individuals and oral malignancy in 1361 individuals [21, 22]. Several international studies indicate that para-habit users are at high risk of OPMDs [29-31]. In Southeast Asia, approximately 90% of deaths resulting from oral malignancy occur in patients with habits known to be associated with enhanced risks of those malignancies; thus, the specific allocation of resources to screen individuals in high-risk groups is justifiable [32]. Due to the often-asymptomatic onset of OPMDs, the majority of patients with OED are evidently unaware of their condition, as was the case in the current study; thus, educating high-risk groups about their risks and encouraging them to engage in the TNOMSP may improve the prevention rates [33].

6.The limitation of the study showed also be mentioned.

Response: We thank reviewer 1 for helping us improve our manuscript. We have revised our conclusion as follows:

Pg. 15

Conclusions

In the current study, a clinical specialist, such as an oral and maxillofacial surgeon well trained and prepared to make a correct diagnosis for OPMDs, demonstrated a better outcome in diagnosing histopathologic OED. Our findings also suggest that, based on the design of the TNOMSP, the high-risk group can be properly referred to diagnosed by the specialist for treatment. However, the limitation of this study is small sample size in a single medical center with limited examiners, but this provides a rationale for further comparative evaluation of the TNOMSP in diagnosis of OPMDs with dysplasia by more qualified examiners or specialists in a larger population base study.

Reviewer #2: In the present paper, the authors investigate the accuracy of diagnoses of OPMDs with dysplasia using Taiwanese Nationwide Oral 6 Mucosal Screening Program. Very interesting work but from reading the abstract and methods section, I was expecting to see some more proposed about the statistical analysis which was claimed by the author. And there are some parts of the research needed to be deliberate.

1. What is the criterion for the receiver operating characteristic (ROC)?

Response: We thank reviewer 2 for the insightful instruction. We apologize for our previous technical mistakes; however, we have uploaded our original data as supplement, and revised table 3-1 and 3-2 to include additional details.

Table 3-1. Accuracy of diagnoses of oral potentially malignant disorders with dysplasia by the primary examiner and the specialist

Patients exclude due to cancer

N = 88 Primary examiner

 OPMDsa

N = 80 Other

N = 8

Pathology report Dysplasia

N = 47 47 (53.4%) 0 (0%)

 Other

N = 41 33 (37.5%) 8 (9.1%)

Sensitivity 100

Specificity 19.5

Positive predictive value 58.8

Negative predictive value 100

Accuracy 62.5

AUC 0.598

a: Oral potentially malignant disorders (OPMDs) include erythroplakia, erythroleukoplakia, non-homogeneous leukoplakia, thick homogeneous leukoplakia, thin homogeneous leukoplakia, verrucous hyperplasia, submucous fibrosis, and lichen planus.

Table 3-2. Accuracy of diagnoses of oral potentially malignant disorders with dysplasia by the specialist

Patients exclude due to cancer

N = 123 Specialist

 OPMDsa

N = 72 Other

N = 51

Pathology report Dysplasia

N = 66 60 (48.8%) 6 (4.9%)

 Other

N = 57 12 (9.8%) 45 (36.6%)

Sensitivity 90.9

Specificity 78.9

Positive predictive value 83.3

Negative predictive value 88.2

Accuracy 85.4

AUC 0.870b

a: Oral Potentially Malignant Disorders (OPMDs) includes erythroplakia, erythroleukoplakia, non-homogeneous leukoplakia, thick homogeneous leukoplakia, thin homogeneous leukoplakia, verrucous hyperplasia, submucous fibrosis, and lichen planus.

b: log-rank test compared with AUC=0.5, p < 0.001

2.The author should deliberate about the description of using the receiver operating characteristic (ROC) curve for sensitivity, specificity, positive predictive value, negative predictive value, and accuracy. Cause there is not enough evidence to support the description of statistical analysis.

Response: We thank Reviewer 2 for this suggestion regarding the ROC curve. We have revised Fig. 3 based on our findings from table 3-1 and 3-2.

Fig. 3 Receiver operating characteristic curve for comparison of the diagnostic accuracy of the primary examiner and the specialist.

The area under the curve (AUC) for the specialist (blue) and primary examiner (green) are 0.870 and 0.598, respectively. The log-rank test comparing the specialist and primary examiner is significant (p < 0.001).

---

## [Editor Report · Decision Letter 1]

16 Dec 2020

Comparison of the accuracy of diagnoses of oral potentially malignant disorders with dysplasia by a general dental clinician and a specialist using the Taiwanese Nationwide Oral Mucosal Screening Program

PONE-D-20-25840R1

Dear Dr. Chen,

We’re pleased to inform you that your manuscript has been judged scientifically suitable for publication and will be formally accepted for publication once it meets all outstanding technical requirements.

Kind regards,

Cheng-Chia Yu

Academic Editor

PLOS ONE
---

## [Editor Report · Acceptance letter]

2 Jan 2021

PONE-D-20-25840R1 

Comparison of the accuracy of diagnoses of oral potentially malignant disorders with dysplasia by a general dental clinician and a specialist using the Taiwanese Nationwide Oral Mucosal Screening Program 

Dear Dr. Chen:

I'm pleased to inform you that your manuscript has been deemed suitable for publication in PLOS ONE. Congratulations! Your manuscript is now with our production department. 

Kind regards, 

on behalf of

Dr. Cheng-Chia Yu 

Academic Editor

PLOS ONE